# Impact of COVID-19 Pandemic on Children and Adolescents with Neuropsychiatric Disorders: Emotional/Behavioral Symptoms and Parental Stress

**DOI:** 10.3390/ijerph19073795

**Published:** 2022-03-23

**Authors:** Francesca Felicia Operto, Giangennaro Coppola, Valentina Vivenzio, Chiara Scuoppo, Chiara Padovano, Valeria de Simone, Rosetta Rinaldi, Gilda Belfiore, Gianpiero Sica, Lucia Morcaldi, Floriana D’Onofrio, Miriam Olivieri, Serena Donadio, Michele Roccella, Marco Carotenuto, Andrea Viggiano, Grazia Maria Giovanna Pastorino

**Affiliations:** 1Child Neuropsychiatry Unit, Department of Medicine, Surgery and Dentistry, University of Salerno, 84081 Salerno, Italy; gcoppola@unisa.it (G.C.); valentina.vivenzio@libero.it (V.V.); chiara.scuoppo@gmail.com (C.S.); chiarapado@hotmail.it (C.P.); valeriades@hotmail.it (V.d.S.); rosetta.rinaldi77@gmail.com (R.R.); gildabelfiore@libero.it (G.B.); luciaa-@hotmail.it (L.M.); floriana.donofrio09@gmail.com (F.D.); mir.olivieri@gmail.com (M.O.); aviggiano@unisa.it (A.V.); graziapastorino@gmail.com (G.M.G.P.); 2Azienda Sanitaria Locale Salerno, Via Nizza 146, 84124 Salerno, Italy; gianpi.sica@gmail.com; 3Department of Psychology, Educational and Science and Human Movement, University of Palermo, 90128 Palermo, Italy; s.donadio1@studenti.unisa.it (S.D.); michele.roccella@unipa.it (M.R.); 4Clinic of Child and Adolescent Neuropsychiatry, Department of Mental Health, Physical and Preventive Medicine, University of Campania “Luigi Vanvitelli”, 80100 Naples, Italy; marco.carotenuto@unicampania.it

**Keywords:** COVID-19, neuropsychiatric disorders, children, adolescents, emotional behavioral symptoms, parental stress

## Abstract

The objective of our study was to evaluate the impact of the COVID-19 pandemic on the emotional and behavioral symptoms in minors with neuropsychiatric disorders and on parental stress through a standardized neuropsychological assessment, comparing the data collected before the pandemic with those collected during the lock-down. Another goal of our study was to analyze the relationship between parental stress and behavioral/emotional symptoms in children. Our study was conducted on 383 families of patients who had already been referred at the Child Neuropsychiatry Unit of the University Hospital of Salerno for different neuropsychiatric conditions. All the parents completed two neuropsychological standardized questionnaires for the assessment of parental stress (PSI—Parenting Stress Index-Short Form) and the emotional/behavioral problems of their children (Child Behaviour CheckList). The data collected during the pandemic were compared with those collected from questionnaires administered during the six months preceding the pandemic, as is our usual clinical practice. The comparison between the mean scores of PSI and CBCL before and after the pandemic showed a statistically significant increase in all subscales analyzed in the total sample. The correlation analysis showed significant positive relationship between the subscale Total Stress of PSI and the subscales Total Problems and Internalizing Problems of CBCL. Our study suggested that the COVID-19 pandemic and the corresponding measures adopted led to an increase in internalizing and externalizing symptoms in children and adolescents with neuropsychiatric disorder. Similarly, parental stress increased during COVID-19 and ahigher level of stress in parents can be related to the internalizing symptoms of their children.

## 1. Introduction

SARS-CoV-2 caused a serious viral disease known as COVID-19. The epidemic, originating in China, was declared a global pandemic on 11 March 2020 [1]. During the first wave, numerous restrictive measures were adopted to contain the infection: the promotion of the use of personal protective equipment (face masks, sanitizing gels), travel restrictions, social distancing and home self-isolation in case of symptoms. In addition, schools, shops, and non-essential production activities were closed, events and ceremonies were canceled, recreational areas were closed, and there were restrictions on exit times for events.

The imposed social isolation, useful for containing the virus, has had, according to numerous studies [2,3], negative effects on the psychological well-being of the general population. The most commonly described negative psychological effects were symptoms of post-traumatic stress, confusion, anxiety, and anger. Stressors included a longer duration of quarantine, fears of getting sick, frustration, boredom, perception of inadequate organization, misinformation, and financial losses. Data on the impact of the health emergency [4] highlight a significant increase in psychopathological discomfort and anxious-depressive symptoms.

The lockdown due to COVID-19 has changed the routines of children and adolescents. Social relationships and school have been replaced by virtual connections and distance learning; according to UNESCO, school closures have limited access to education to over 1.5 million children and young people in 165 countries. Some of the possible negative consequences on children and adolescent mental health, caused by distance learning, are increasing inequality among families due to different availability of digital resources [5], and reduction of physical activity and social interactions [6]. In Italy, the closure of schools during the lockdown caused a disservice in the daily life of millions of children and adolescents, who make up about 16% of the Italian population.

Specialists were concerned with the long-term psychological impact of the pandemic and containment measures on children and adolescents. In a study conducted in Italy and Spain [7], a worsening of the emotional state and behavior of children was highlighted, and difficulties in attention and concentration, boredom, irritability, and feelings of loneliness are reported. Italian children are perceived by their parents as sadder than their Spanish peers, probably because of a longer lasting quarantine.

Some studies conducted in China [8,9] related to the consequences of COVID-19 highlighted that the prevalence of depressive symptoms in primary and secondary school students had widely increased. A survey conducted by Spinelli et al. documented how the increase in parental stress levels related to the fear of health risks affected the well-being of children [10].

The quarantine, in fact, has had traumatic consequences not only on the adult population, but also on the pediatric population [11]. Publications [12,13] have shown that quarantine and social isolation have had a negative effect on the psychological well-being of children and adolescents, especially in cases of pre-existing conditions; subjects with psychopathologies or neurodevelopmental disorders already present are mainly affected.

There are not many studies in the literature that have evaluated the effects of restrictions on a considerable sample of patients referred to child neuropsychiatry services, and compared emotional–behavioral problems and parental stress in the pre- and pandemic periods.

In the study by Sesso et al. [14] a positive correlation was found between parental stress and the emotional–behavioral problems of children during the period of the lockdown due to COVID-19, particularly in the neuropsychiatric population.

Studies [15,16] conducted on patients referred to the child neurology and neuropsychiatry services showed a global worsening of behavioral symptoms, reduced compliance with distance learning and a progressive change in daily life. In addition, an increase in early childhood somatic and anxiety symptoms was highlighted. Moreover, an increase in behavioral problems and post-traumatic symptoms in older children and adolescents was found.

In a study conducted by Zhang et al. [17] on children with Attention/Deficit Hyperactivity Disorder (ADHD) during the COVID-19 epidemic, a worsening of inattention and hyperactivity symptoms, stress and mood disorders was highlighted.

Regarding epileptic conditions, researchers [18,19] considered an increase in behavioral problems and indicated that the pandemic and associated restrictions have had a negative impact on young people with epilepsy. In a study by Wanigasinghe et al. [20], it emerged that parents of children with epilepsy who adopted healthy management strategies during the COVID-19 lockdown did not experience high levels of stress.

In a study of subjects with intellectual disabilities by Bailey et al. [21], no difference was found in children’s behavior and emotional problems during the COVID-19 restrictions.

Regarding Autism Spectrum Disorder (ASD), the study by Alhuzimi [22] found that, during the lockdown, familiar status had a significant impact on parental stress and on emotional well-being. Furthermore, parental stress and emotional–behavioral problems were negatively impacted by the frequency and usefulness of the support received during the pandemic. These were also adversely affected by the change in the severity of ASD behaviors of children. Children with ASD seem to bea particularly vulnerable population during the COVID-19 pandemic due to the potential exacerbation of ASD symptoms, limited access to therapies, and the heavy responsibility placed on parents. In the study by Theis et al. [23], over 90% of parents of children and adolescents with intellectual disabilities reported a negative impact on mental health, including behavioral problems, mood disorders, social regression and learning difficulties.

Italian studies [14,24] on children and adolescents with psychiatric disorders found a worsening of psychological distress in patients with externalizing behavioral symptoms, while patients with internalizing disorders showed a better adaptation to the pandemic context. Furthermore, there was a significant positive correlation between internalizing symptoms of children and parental stress.

The objective of our study was to evaluate the impact of the COVID-19 emergency and the related measures adopted (closure of schools, closure of rehabilitation centers, traffic ban, etc.) on the emotional and behavioral symptoms in minors with neuropsychiatric disorders and on parental stress through a standardized neuropsychological assessment. Another goal of our study was to analyze the relationship between parental stress and behavioral/emotional symptoms in children.

## 2. Materials and Methods

### 2.1. Participants

Our observational study was conducted on 383 families of patients who were already referred to the Child Neuropsychiatry Unity of the University Hospital of Salerno. The inclusion criteria were as follows: (i) age under 18 years, (ii) presence of a neuropsychiatric disorder assessed in the Child Neuropsychiatry Unity of the University Hospital of Salerno by a multidisciplinary team composed of child neuropsychiatrists, psychologists and speech therapists and by standardized tests (Autism Diagnostic Observation Schedule—Second Edition (ADOS-2), Autism Diagnostic Interview—Revised (ADI-R), Kiddie Schedule for Affective Disorders and Schizophrenia—Present and Lifetime (K-SADS-PL), Child Behavior Checklist (CBCL), Wechsler Intelligence Scales for Children (WISC), Conners, etc.) in the period between the years 2018 and 2019.

The data collection was carried out in the months of March–May 2020 by six psychologists and two child neuropsychiatrists. In a first phase, we contacted the parents by phone and administered a telephone interview for a duration of about 20 min. In a second phase, parents were asked to complete two neuropsychological standardized questionnaires for the assessment of parental stress (PSI—Parenting Stress Index-Short Form) and the emotional–behavioral problems of their children (Child Behavior CheckList). This questionnaire was sent by e-mail; alternatively, parents could fill in the questionnaire through an online program created ad hoc.

The data collected during the pandemic were compared with that collected from questionnaires administered in the six months preceding the pandemic, as is usual clinical practice.

All participants were given a detailed explanation of the purpose and the procedures of the study. Parents provided their informed consent in written form or by e-mail. The procedure was approved by the Campania South Ethics Committee (protocol number N.0061902, 2 April 2020) and was conducted according to the rules of good clinical practice in line with the Helsinki Declaration.

### 2.2. Telephone Interview

The qualitative telephone interview was based on a series of questions specially created by the authors to collect general demographic information (for example, age of parents and their level of education) and to investigate if the changes in life habits related to lock-down had had an impact on the psycho-physical well-being of their children and family (for example, type of diet, sleep patterns, increase in problem behaviors, etc.). More specifically, the interview included the following questions:-In your opinion, family management during the lockdown period worsened, was unchanged or improved?-If family management has worsened, what are the principal reasons that led to difficulties in management?-In your opinion, did the symptoms related to your child’s medical condition worsen during the lockdown period?-In your opinion, is your child having sleep problems that were not present before?-In your opinion, is your child having eating problems that were not present before?-On a scale of 1 to 10, how concerned are you about the general pandemic situation?-What are your main reasons for concern?-Are you benefiting or have you benefited from online psychological support provided for the COVID-19 emergency?-What are the reasons why you did not benefit from online psychological support?

### 2.3. Child Behavior CheckList

The Child Behavior Checklist (CBCL) is an evidence-based questionnaire [25] used to assess behavioral, emotional and social problems and functioning in children up to 18 years. There are two versions compiled by the caregivers: one is used for children from 1 and a half to 5 years old and another from 6 to 18 years old. The tool consists of 113 statements and there are three possible answers recorded on a Likert scale: 0 Not true, 1 Sometimes or Fairly True, 2 Often True or Very True. The results are distributed in subscales as t-scores. The normative data are divided as follows: a t-score ≤64 is normal, an interval at the limits is indicated by a t-score between 65 and 69, and a t-score ≥70 indicates clinical symptoms. For the subscales of the “internalization problems”, “externalization problems” and “total problems”, a t-score ≤59 indicates normal scores, a t-score between 60 and 64 indicates a score that is within a boundary range, and high levels of maladaptive behavior are indicated by a t-score ≥65. The CBCL have an adequate internal consistency, as emerged from the study of D’Orlando et al. [26]. Coefficient alpha reliability coefficients based on the responses of individuals in the normative sample ranged from 0.66 to 0.95 for 1 and a half to 5 years subscales and from 0.72 to 0.97 for 6 to 18 years subscales. Test–retest reliability score ranged from 0.68 to 0.92 for 1 and a half to 5 years subscales and from 0.82 to 0.94 for 6 to 18 years subscales.

### 2.4. Parental Stress Index

The PSI Short Form (PSI/SF) arises from the Parenting Complete Stress Index (PSI) test [27] and is composed of 36 sentences. The parents must indicate how much they agree with each of the 36 sentences, according to a 5-point Likert scale, ranging from “strongly agree” to “strongly disagree”. In this questionnaire there are three subscales: Parental Distress (PD), which indicates the level of discomfort that a caregiver is experiencing as a parent, also taking into consideration personal factors directly related to that role, Dysfunctional Parent-Child Interaction Scale (P-CDI), which assesses the level of satisfaction linked to the relationship with the child, and finally Difficult Child Scale (DC), which assesses the parent’s perception of having a difficult child [28]. In PSI/SF the raw score is converted to t-scores; the higher the t-score the higher the stress levels, and in particular a t-score ≥85 indicates clinically significant parental stress [27]. The test also includes a defensive response scale (DF) useful for verifying the validity of the protocol as it indicates whether the parent tends, for example, to present a better self-image or to minimize problems and perceived stress in the relationship with the child.

Coefficient alpha reliability coefficients based on the responses of individuals in the normative sample ranged from 0.78 to 0.88 for Child Domain subscales and from 0.75 to 0.87 for Parent Domain subscales. Reliability coefficients for the two domains and the Total Stress scale were 0.96 or greater, indicating a high degree of internal consistency for these measures. Test–retest reliability coefficients, obtained through several studies, ranged from 0.55 to 0.82 for the Child Domain, from 0.69 to 0.91 for the Parent Domain, and from 0.65 to 0.96 for the Total Stress score.

### 2.5. Statistical Analysis

All neuropsychological scores were expressed as mean ± standard deviation (SD). In order to verify the data distribution, the Kolmogorov-Smirnov normality test was preliminarily performed. Because of the presence of some data not normally distributed, non-parametric methods were employed for our analysis. The Wilcoxon signed-rank test was performed for the mean score comparison in paired samples. The Spearman correlation test was employed in order to evaluate the relationship between different variables. All data were analyzed using the Statistical Package for Social Science software, version 23.0 (IBM Corp., Armonk, NY, USA, 2015); a *p* value <0.05 is considered statistically significant.

## 3. Results

### 3.1. Background

This study has been conducted on a population residing in Campania, a region of southern Italy. The data collection took place over a period of time between 23 March 2020 and 15 May 2020, during which the Italian government imposed a total lockdown on the entire national territory.

In fact, in February 2020, Codogno, a city in Lombardia in northern Italy, became the epicenter of COVID-19 in Europe. From this moment on, the Italian government implemented a series of gradually more restrictive measures with the aim of avoiding a rapid spread of the infection throughout the national territory.

The progressive measures adopted by the government through the legislative decrees of 9, 11 and 22 March 2020 were:-Obligation for everybody to remain in their home;-Limitation of the free circulation of people (only allowed to deal with necessary situations, after filling a self-certification form);-Mandatory use of a mask to cover nose and mouth outside one’s own home;-Prohibition on gatherings, with an obligation to respect the safety distances between one individual and another (at least 1 m);-Cancellation of all face-to-face teaching activities (including exams) from kindergartens to universities and the beginning of distance learning;-Closure of non-residential rehabilitation centers;-The ceasing of all productive activities not considered indispensable;-Promotion of working from home;-Closure of restaurants and shops, an exception made for those that trade essential goods such as food, pharmacies, newsagents, tobacco shops, electronics stores, hardware stores, opticians, laundries and funeral homes;-Closing of beauty centers, hairdressers, barbers etc.; only services guaranteed were those of banks, post offices and insurance companies;-Closure of all recreational and sporting activities (cinemas, theatres, museums, gyms, etc.);-Suspension of religious celebrations.

The rigid nature of the measures adopted in Italy to contain the spread of the Coronavirus was a severe blow to the Italian population both from an economic point of view, with the closure of almost all production and commercial activities, and from a psychological point of view, due to the fear of contagion but also to the drastic change in life habits, the limitations on personal freedom and the drastic reduction in interpersonal relationships.

### 3.2. Sample Characteristics

The total sample included 383 families. The families which were initially invited to take part in the study numbered 398, but 15 (3.9%) of these chose not to participate. The children was aged between 2–18 years (mean age = 9.89 ± 4.42; male = 233, 61%). The total sample included children and adolescents with the following neuropsychiatric diagnosis: autism spectrum disorder (*n* = 114), epilepsy (*n* = 93), specific learning disorders (*n* = 41), intellectual disability (*n* = 34), communication disorders (*n* = 32), attention deficit/hyperactivity disorder (*n* = 21), behavioral disorders (*n* = 16), anxiety disorders (*n* = 16), mood disorders (*n* = 16). In Table 1 and Table 2 are summarized the main socio-demographic and clinical characteristics. In Table 3 the principal questions in the telephone interview are reported.

### 3.3. Comparison of Mean Scores at PSI and CBCL before–during COVID-19 Pandemic

The comparison between mean PSI scores before and after the pandemic showed a statistically significant increase in all subscales analyzed in the total sample (*p* < 0.05 in all the PSI subscales).

Similarly, the mean scores of all CBCL subscales were statistically increased during the pandemic compared to the pre-pandemic period (*p* < 0.05 in all the CBCL subscales).

The mean PSI and CBCL scores in the total sample and the statistical comparison are shown in Table 4.

In the subsample with autism all the subscales of PSI and CBCL were significantly increased (*p* < 0.05) except attention problems (Z = −1.851, *p* = 0.064) and pervasive problems (Z = −1.450, *p* = 0.147).

In the subsample with epilepsy and specific learning disorder, all the subscales of PSI and CBCL were significantly increased (*p* < 0.05). In the intellectual disability subsample, all the subscales of PSI and CBCL were significantly increased (*p* < 0.05), except emotional response (Z = −1.823, *p* = 0.068), withdrawal (Z = −1.602, *p* = 0.109), attention problems (Z = −1.684, *p* = 0.092) and pervasive problems (Z = −1.682, *p* = 0.092). In the subsample with communication disorder, there were no significant differences in the PSI subscales, while the average scores of the following CBCL scales were significantly increased: anxiety/depression (Z = −2.360, *p* = 0.018), social problems (Z = −1.962, *p* = 0.050), aggressive problems (Z = −2.317, *p* = 0.020), affective problems (Z = −2.002, *p* = 0.045), anxiety problems (Z = −2.180, *p* = 0.029), emotional response (Z = −2.289, *p* = 0.022), anxiety/depression (Z = −1.994, *p* = 0.046). In the subsample with attention deficit/hyperactivity disorder, there were no significant differences in the PSI subscales, while the average scores of the following CBCL scales were significantly increased: anxiety/depression (Z = −2.246, *p* = 0.025), somatic complaints (Z = −2759, *p* = 0.006), rule breaking behavior (Z = −2.096, *p* = 0.036), internalizing problems (Z = −2.115, *p* = 0.034), affective problems (Z = −2.615, *p* = 0.009), anxiety problems (Z = −2.187, *p* = 0.029), somatic problems (Z = −2.441, *p* = 0.015), conduct problems (Z = −2.226, *p* = 0.026).

In the subsample with behavioral disorder, all the subscales of PSI and CBCL were significantly increased (*p* < 0.05). In the subsample with anxiety, all the subscales of PSI and CBCL were significantly increased (*p* < 0.05) except difficult child (Z = −1.774, *p* = 0.760) and thought problems (Z = −1.855, *p* = 0.064). In the subsample with mood disorders, all the subscales of PSI and CBCL were significantly increased (*p* < 0.05) except parental distress (Z = −1.334, *p* = 0.182), difficult child (Z = −1.102, *p* = 0.271), total stress (Z = −1.481, *p* = 0.139) and thought problems (Z = −0.612, *p* = 0.541).

### 3.4. Correlation Analysis between PSI and CBCL Subscales

The correlation analysis showed significant positive relationship between the subscales Total Stress (PSI) and Total Problems (CBCL) and Internalizing Problems (CBCL). A positive significant relationship was also found between the subscale Parent–Child Difficult Interaction (PSI) and Total Problems (CBCL) and Internalizing Problems (CBCL). Finally, we found a significant positive relationship between the subscale Difficult Child (PSI) and the subscale Externalizing Problems (CBCL). All the results are summarized in Table 5.

## 4. Discussion

The aim of our study was to evaluate the impact of the COVID-19 pandemic on children and adolescents with neuropsychiatric disorders, comparing the pre-pandemic period with the first months of lockdown. Our comparison was based on standardized neuropsychological tests that evaluated emotional/behavioral symptoms (CBCL) in children and parental stress (PSI) level in their parents. Our clinical sample included the families of children diagnosed with Autism Spectrum Disorder, Specific learning Disorder, Attention Deficit/Hyperactivity Disorder, Communication Disorder, Intellectual Disability, Anxiety Disorder, Depressive Disorder, Behavioral Disorder and Epilepsy.

Overall, our findings showed that lockdown had a psychological impact on children with neurodevelopmental disorders and their families. The analysis of the CBCL results showed a worsening in both internalizing and externalizing symptoms during the lockdown period in the entire sample. This result is consistent with the literature: in a study by Conti et al. [16], the CBCL of 141 subjects under the age of 18 with Neurologic and Psychiatric disorders were compared, and a worsening in emotional and behavioral problems emerged during the lockdown period. It could be hypothesized that confinement in the home and the loss of daily rhythms increase the emotional and behavioral problems in these children because they have more difficulty in adapting to new situations.

In our study, in particular, children with ASD aged ≥6 years had a significant increase in all externalizing, internalizing, and total problems. These results, in agreement with the literature [16], suggested that lockdown, cessation of therapies and school closures negatively impacted the psychological well-being of these subjects. Our results also showed that in children with ASD under the age of 6 there was worsening in all areas except attention and pervasive developmental problems. In our sample, these children may show a worsening in different behavioral and emotional areas because of the lack of distance learning as alternative to face-to-face schooling, leading to the children not having activities to do during the day, which could promote possible higher levels of stress and emotional dysregulation.

In children with ADHD there was a significant increase in levels of anxiety/depression, somatic disorders, rule violation, affective disorders, anxiety disorders, somatic disorders and conduct disorders. Our finding is in agreement with a systematic review by Behrmann [29], which showed that the impact of the pandemic was generally negative for individuals with ADHD who experienced behavioral and psychological issues, such as depression, anxiety, loneliness, boredom and emotional distress, particularly in adolescents and preteens.

In the Epilepsy and SLD groups, we found significantly higher scores in all the CBCL areas during the pandemic. Our results agree with the literature: in some studies [19,30] a worsening of the emotional state in children with epilepsy emerged; these findings suggest that the pandemic and associated restrictions have had a negative impact on young people with epilepsy. As for subjects with SLD, our results are in agreement with a study by Soriano-Ferre et al. [31] who showed that children with dyslexia aged 9 to 14 had increased levels of depression and anxiety symptoms during quarantine.

In the anxiety group and mood disorders groups, all the emotional–behavioral symptoms analyzed were significantly worsened except thought disorders. Stressful factors such as having to stay indoors, the fear of contracting the disease and the loss of social relationships negatively affect children and adolescents without psychological problems, leading to an increase in anxiety, depression and irritability [32]. Therefore, children with pre-existing mood and anxiety problems could have a higher probability of the worsening of their emotional symptoms.

The subjects with ID presented more emotional and behavioral problems during the lockdown in all areas explored, except emotional reactivity, attention, withdrawal and pervasive developmental problems, which did not show significant worsening compared to the pre-emergence period. Our results are in agreement with the study by Bailey et al. [21] in which no difference was found in children’s behavior and emotional problems, suggesting that the restrictions and concerns concerning COVID-19 took longer to affect these families. As far as we know, this is the only study comparing data pre- and during the lockdown period in children with DI.

An increase in externalizing and internalizing problems emerged in the behavioral disorders group and in the communication disorder group. There do not seem to be any studies exploring emotional and behavioral problems in subjects with communication and behavioral disorders during the lockdown. It is important to highlight that the restrictive measures and the interruption of routines led to an increase of emotional and behavioral difficulties in our entire sample, regardless of the diagnosis, probably because the neuropsychiatric pediatric population is more vulnerable than the general population.

It is already known in the literature that there is a correlation between neurodevelopmental disorders and increased parental stress [30,33,34,35,36]. Regarding parental stress levels during a lockdown period, the scores for PSI were significantly higher in all clinical groups. We can hypothesize, that due to home confinement, parents’ difficulties in managing their child with a psychiatric problem have increased: the reduced presence of therapists, relatives or other support figures could increase stress levels. In fact, in telephone interviews with parents, changes in daily routine, home confinement, distance learning and discontinuity in therapies are confirmed among the reported main causes of stress. Additionally, children can experience stress as they stay at home for a long time rather than at school and this can negatively affect how they interact with their siblings and parents, increasing the overall stress levels of all family members.

Analyzingthe PSI results in more detail, in the group of subjects with anxiety disorders, only Parental Distress (PD), Total Stress (TS), and Parent–Child Dysfunctional Interaction (P-CDI) scores were significantly higher during the pandemic period. This result suggested that, in this group, the perception of having a complicated relationship with sons was significantly increased in the post-pandemic period, revealing a greater perception of parental role-related stress and total stress. Instead, parents did not feel their children as more problematic than in the pre-pandemic period. In the group of subjects with depression disorders, the Parent–Child Dysfunctional Interaction (P–CDI) area was significantly higher, indicating greater difficulty in the parent–child relationship. Finally, the group of subjects with communication disorders was the only one that did not show significant differences in the perception of stress during the pandemic period compared to the pre-pandemic.

In our study a significant relationship was found between some CBCL and PSI subscales. The correlation between internalizing problems and parental stress can be related to a significant difficulty of parents in understanding the symptoms of their children, which brings them a sense of helplessness. Parental stress in our sample was positively related with emotional and behavioral symptoms in children, in particular with internalizing problems. This result agrees with the study by Sesso et al. [15] conducted on 77 subjects with neurodevelopmental disorders. The authors showed a significant correlation between a worsening of internalizing symptoms and the level of parental stress during the lockdown period. In our sample, internalizing problems were also related to a difficulty in parent–child interaction. We can therefore hypothesize that internalizing symptoms, such as anxiety and mood disorders, can lead to an emotional withdrawal of the children that affects the intra-family relationship. On the other hand, we can hypothesize that internalizing symptoms are more difficult to recognize by parents than externalizing ones, so they can go unnoticed and worsen over time. Finally, as might be expected, externalizing problems were more related to the parental perception of a difficulty due to the child’s own characteristics, rather than a difficulty in their parental role.

The main strengths of our study are the large sample size and the data comparison between pre- and during lockdown period, which allows us to analyze the change occurring in this time frame. The main limitations are possible selection bias, inhomogeneity of the sample and the difficulty of some families in completing the online questionnaires. Another important limitation is that we considered only the first months of the COVID-19 pandemic, for this reason we intend; future studies should analyze the emotional/behavioral symptoms and parental stress during the months following the pandemic, and compare the changes over time.

In conclusion. our data show how the restrictions due to the pandemic have negatively impacted the psychological well-being of families; in fact, emotional and behavioral symptoms in children have arisen and parent–child interaction and family management seem more difficult. Parents have had to deal with their children’s problems without outside help, such as from school or therapy, and this has made them more concerned and has diminished their sense of effectiveness. Our results, together with those in the literature, suggest that we need to monitor the psychological well-being of young people with neurodevelopmental disorders and of their families in a period of pandemic emergency.

## Figures and Tables

**Table 1 ijerph-19-03795-t001:** Main socio-demographic characteristics of the families that took part in our study and of families who refused to participate. SD = standard deviation. * calculated by school age.

**Participants *n* = 383**	
child age (mean ± SD)	9.89 ± 4.42
sex	
male	233 (61%)
female	150 (39%)
father age (mean ± SD)	43.87 ± 7.05
mother age (mean ± SD)	40.82 ± 6.34
maternal education level (mean ± SD) *	14.07 ± 3.89
paternal education level (mean ± SD) *	14.07 ± 3.71
**Families who refused to participate in the study *n* = 15**	
child age (mean ± SD)	10.73 ± 4.48
sex	
male	9/15(%)
female	6/15(%)
father age (mean ± SD)	43.40 ± 2.64
mother age (mean ± SD)	39.67 ± 2.06
maternal education level (mean ± SD) *	14.27 ± 3.15
paternal education level (mean ± SD) *	13.73 ± 3.24
Diagnosis	Autism Spectrum Disorder = 4 Epilepsy = 3 Specific Learning Disorder = 2 Intellectual Disability =3

**Table 2 ijerph-19-03795-t002:** Main socio-demographic and clinical characteristics of the total sample and of the sub-samples divided by main neuropsychiatric diagnoses. * The neuropsychiatric diagnoses were made according to the diagnostic criteria of the Diagnostic and Statistical Manual of Mental Disorders 5th edition (DSM-5).

Principal Diagnosis *	Age (Years)	Sex	Father Age (Years)	Mother Age (Years)	Characteristics of the Principal Disease	Neuro-Psychiatric Comorbidities	Other Clinical Conditions	Drug Therapy
autism spectrum disorder *n* = 114	8.1 ± 4.24	male = 79 (69%)	45.18 ± 6.74	41.81 ± 6.32	level 1 = 44 (39%) level 2 = 43 (38%) level 3 = 27 (24%)	35 (30%)	29 (25%)	22 (19%)
epilepsy *n* = 93	13.50 ± 7.78	male = 53 (57%)	42.50 ± 10.61	37.50 ± 10.61	focal = 51 generalized = 31 unknown = 11	26 (28%)	17 (18%)	82 (88%)
specific learning disorders *n* = 41	10.50 ± 2.11	male = 22 (54%)	39.72 ± 4.51	38.00 ± 3.49	mixed = 28 (68%) dyslexia + dysorthography = 10 (27%) only dyscalculia = 3 (7%)	7 (17%)	7 (17%)	0 (0%)
intellectual disability *n* = 34	9.00 ± 4.93	male = 18 (53%)	43.42 ± 6.21	40.41 ± 6.33	mild = 21 (62%) moderate = 9 (29) severe = 4 (12%)	13 (38%)	14 (41%)	7 (21%)
communication disorders *n* = 32	5.09 ± 1.44	male = 20 (63%)	38.23 ± 7.17	35.56 ± 5.03	language disorder = 20 (63%) speech sound disorder = 10 (31%) stuttering = 2 (6%)	4 (13%)	5 (15%)	0 (0%)
attention deficit/hyperactivity disorder *n* = 21	10.76 ± 3.51	male = 15 (71%)	44.33 ± 8.12	40.52 ± 6.48	combined = 15 (71%) inattentive = 3 (19) hyperactive/impulsive = 2 (10%)	5 (24%)	3 (14%)	12 (57%)
behavioral disorders *n* = 16	11.67 ± 2.58	male = 12 (75%)	42.50 ± 4.52	40.13 ± 4.40	oppositional-defiant disorder = 13 (81%) conduct disorder = 3 (19%)	12 (75%)	2 (13%)	7 (44%)
anxiety disorders *n* = 16	11.34 ± 3.45	male = 7 (44%)	44.44 ± 7.74	43.00 ± 5.59	generalized anxiety = 9 (56%) school phobia = 5 (31%) social anxiety disorder = 3 (19%)	12 (75%)	2 (13%)	12 (75%)
mood disorders *n* = 16	12.50 ± 2.79	male = 7 (44%)	43.00 ± 6.70	40.06 ± 4.92	depressive disorder = 11 (69%) bipolar disorder = 5 (31%)	12 (75%)	2 (13%)	14 (88%)
Total sample *n* = 383	9.89 ± 4.42	male = 233 (61%)	43.87 ± 7.05	40.82 ± 6.34	-	126 (33%)	81 (21%)	156 (41%)

**Table 3 ijerph-19-03795-t003:** Parents’ responses to the telephone interview.

Questions	
In your opinion, the family management during this lockdown period is:	
worse	275/383 (71.80%)
unchanged	89/383 (23.23%)
improved	19/383 (4.96%)
If the family management has worsened, what are the principal reasons that led to difficulties in management? (more than one answer can be provided)	
changes in daily routine	195/275 (70.90%)
home confining	185/275 (67.27%)
distance learning	178/275 (64.72%)
discontinuation of rehabilitative therapy	146/275 (53.09%)
decrease in social relationships	113/275 (41.09%)
emotional impact linked to the perception of danger	98/275 (35.63%)
Smart-working	71/275 (25.81%)
In your opinion, the symptoms related to your child’s medical condition worsened during the lockdown period.	
yes	168/383 (43.86%)
no	215/383 (56.14%)
In your opinion, is your child having sleep problems that were not present before?	
yes	114/383 (29.77%)
no	269/383 (70.23%)
In your opinion, is your child having eating problems that were not present before?	
yes	108/383 (28.20%)
no	275/383 (71.80%)
On a scale of 1 to 10, how concerned are you about the general pandemic situation?	7.6 ± 2.13 (mean ± standard deviation)
What are your main reasons for concern? (more than one answer can be provided)	
fear of COVID contagion	182/383 (47.51%)
worsening of the child’s clinical symptoms	118/383 (30.80%)
difficulty in obtaining medical assistance	84/383 (21.93%)
difficulties in family routine and management	78/383 (20.36%)
emergence or worsening of emotional/behavioral problems in the child	58/383 (15.14%)
Are you benefiting or have you benefitedfrom online psychological support providedduring the COVID-19 emergency?	
yes	8/383 (2.08%)
no	375/383 (97.91%)
What are the reasons why you did not benefitfrom online psychological support?	
I was not aware of the online psychological support service	178/375 (47.46%)
I didn’t consider online psychological support necessary	129/375 (34.40%)
Online psychological support was not provided in my municipality	39/375 (10.41%)
Other	29/375 (7.73%)

**Table 4 ijerph-19-03795-t004:** Statistical comparison between average scores of the Child Behavior CheckList and the Parental Stress Index at time 0 (before the pandemic) and at time 1 (March–May 2020).

Standardized Neuropsychological Test	Time 0 (Mean ± SD)	Time 1 (Mean ± SD)	Statistic (Wilcoxon Test)	*p* Value	Ꞃ^2^	DCohen
**Parental Stress Index (PSI)** ***n* = 383**						
Parental Distress (PD)	60.29 ± 27.27	72.38 ± 28.04	Z = −10.540	** *p* ** **< 0.001**	0.290	1.278
Parent–Child Difficult Interaction (*p*-CDI)	63.61 ± 24.29	76.82 ± 24.65	Z = −11.318	** *p* ** **< 0.001**	0.334	1.418
Difficult Child (DC)	63.84 ± 26.57	75.99 ± 26.47	Z = −10.379	***p* < 0.001**	0.281	1.251
Total Stress (TS	63.45 ± 24.63	76.87 ± 25.94	Z = −12.281	** *p* ** **< 0.001**	0.394	1.612
**Child Behavior CheckList (CBCL)** **6–18 years** ***n* = 315**						
Anxiety/Depression	59.36 ± 7.53	68.76 ± 13.18	Z = −11.773	** *p* ** **< 0.001**	0.440	1.773
Withdrawal/Depression	61.55 ± 8.49	69.44 ± 12.73	Z = −10.652	** *p* ** **< 0.001**	0.360	1.501
Somatic complaints	58.52 ± 7.97	68.59 ± 13.43	Z = −10.879	** *p* ** **< 0.001**	0.376	1.552
Socialization	62.20 ± 8.74	69.77 ± 12.92	Z = −10.206	** *p* ** **< 0.001**	0.331	1.406
Thought problems	61.40 ± 9.87	66.06 ± 11.96	Z = −7.993	***p* < 0.001**	0.203	1.009
Attention problems	62.75 ± 9.40	68.88 ± 11.48	Z = −10.109	** *p* ** **< 0.001**	0.324	1.386
Rule-breaking behavior	58.16 ± 7.02	65.28 ± 12.59	Z = −10.396	** *p* ** **< 0.001**	0.343	1.445
Aggressive behavior	60.80 ± 10.40	67.13 ± 13.26	Z = −10.110	** *p* ** **< 0.001**	0.324	1.386
Affective problems	61.87 ± 8.14	68.71 ± 11.92	Z = −11.132	** *p* ** **< 0.001**	0.326	1.391
Anxiety problems	61.64 ± 7.58	69.26 ± 11.95	Z = −10.989	** *p* ** **< 0.001**	0.383	1.577
Somatic Problems	57.11 ± 8.12	64.39 ± 11.54	Z = −9.823	***p* < 0.001**	0.306	1.329
ADHD	60.52 ± 7.36	66.23 ± 10.07	Z = −9.959	***p* < 0.001**	0.315	1.356
Oppositional-defiantproblems	58.01 ± 7.46	64.02 ± 10.93	Z = −9.950	***p* < 0.001**	0.314	1.354
Conduct problems	57.05 ± 7.03	62.92 ± 10.67	Z = −10.769	** *p* ** **< 0.001**	0.368	1.527
Internalizing problems	59.86 ± 9.95	68.37 ± 13.37	Z = −11.227	** *p* ** **< 0.001**	0.400	1.633
Externalizing problems	57.69 ± 9.72	65.83 ± 14.03	Z = −10.614	** *p* ** **< 0.001**	0.358	1.492
Total Problem	60.41 ± 9.53	68.77 ± 13.71	Z = −11.757	** *p* ** **< 0.001**	0.439	1.769
**Child Behavior CheckList (CBCL)** **1.5–5 years** ***n* = 68**						
Emotional response	54.54 ± 11.71	66.40 ± 13.63	Z = −4.518	** *p* ** **< 0.001**	0.300	1.310
Anxiety/Depression	55.69 ± 8.56	64.21 ± 12.73	Z = −5.026	** *p* ** **< 0.001**	0.371	1.538
Somatic complaints	55.03 ± 7.08	63.29 ± 12.47	Z = −4.680	** *p* ** **< 0.001**	0.322	1.379
Withdrawal	63.62 ± 12.94	68.56 ± 14.42	Z = −2.743	** *p* ** **= 0.006**	0.111	0.705
Sleep problems	56.91 ± 11.10	59.49 ± 11.49	Z = −3.201	** *p* ** **= 0.001**	0.151	0.842
Attention problems	61.93 ± 10.63	65.00 ± 12.35	Z = −2.436	** *p* ** **= 0.015**	0.087	0.618
Aggressive behavior	56.22 ± 9.32	66.34 ± 15.30	Z = −5.078	** *p* ** **< 0.001**	0.379	1.563
Affective problems	56.63 ± 9.00	63.19 ± 12.50	Z = −4.063	** *p* ** **< 0.001**	0.243	1.132
Anxiety problems	56.54 ± 8.32	63.69 ± 12.12	Z = −4.337	** *p* ** **< 0.001**	0.277	1.237
Pervasive Problems	63.74 ± 12.16	66.29 ± 12.54	Z = −2.008	** *p* ** **= 0.045**	0.059	0.502
ADHD	59.26 ± 8.39	62.94 ± 9.11	Z = −3.313	** *p* ** **= 0.001**	0.161	0.877
Oppositional-defiant problems	59.09 ± 7.24	59.96 ± 9.76	Z = −4.186	** *p* ** **< 0.001**	0.258	1.178
Internalizing problems	56.71 ± 12.50	64.44 ± 15.86	Z = −4.308	** *p* ** **< 0.001**	0.273	1.225
Externalizing problems	56.04 ± 12.11	65.66 ± 17.03	Z = −4.508	** *p* ** **< 0.001**	0.299	1.306
Total Problems	57.69 ± 13.55	65.56 ± 16.92	Z = −3.999	** *p* ** **< 0.001**	0.235	1.109

*p* values < 0.05 are in bold.

**Table 5 ijerph-19-03795-t005:** Spearman correlation analysis between Child Behavior Checklist and Parental Stress Index subscales. CBCL= Child Behavior Checklist.

			CBCL Total Problems	CBCL Externalizing Problems	CBCL Internalizing Problems
**Parental** **stress**	**Parental Distress**	*r*	0.081	0.042	0.062
		*p* value	0.112	0.415	0.230
	**Parent–Child Difficult Interaction**	*r*	**0.115**	0.070	**0.102**
		*p* value	**0.024**	0.172	**0.045**
	**Difficult Child**	*r*	0.089	**0.128**	0.092
		*p* value	0.082	**0.012**	0.071
	**Total Stress**	*r*	**0.107**	0.075	**0.103**
		*p* value	**0.036**	0.144	**0.043**

*p* value < 0.05 and *r* > 0.1 are in bold.

## Data Availability

The original data that support the findings of this study are available from the corresponding author.

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
