# Peer review of "Impact of COVID-19 Pandemic on Children and Adolescents with Neuropsychiatric Disorders: Emotional/Behavioral Symptoms and Parental Stress"

_ijerph, 2022, doi:10.3390/ijerph19073795_

Round 1

Reviewer 1 Report

The present manuscript deals with the important actual topic, impact of Covid-19 pandemic on children and adolescents with neuropsychiatric disorders. The authors argue that while there is already much work on the impact of a pandemic on mental health, the impact of pandemic on children with neuropsychiatric disorders has not been adequately explored. The study may provide useful findings, but it should need several improvements.

Abstract:

I recommend not including the words "Introduction", "Material and Methods", "Results", and "Conclusions". Although abstracts usually have a similar structure, it is not common to include these designations to text.

Materials and Methods:

„Our observational longitudinal study…“  The study was not longitudinal, only it compared two datasets within a period of six months.

In the abstract the authors stated that „the data collected during the pandemic were compared with that collected from questionnaires administered in the six months preceding the pandemic“, however, on page 4 they stated that „ The data collected during the pandemic were compared with that collected from questionnaires administered in the three months preceding the pandemic, as usual clinical practice.“ Thus, this contradiction needs a clarification.

We know that the pandemic has been going on for two years. Long lockdowns caused many mental problems. However, the data from the present study was gathered in March-April 2020, thus, just at the beginning of a pandemic. Thus, they are not very representative, which should be stated in text discussion, et least. The pandemic had several stages, alternating releases with lockdowns, in different parts of the world at different times, and the effects of the pandemic were different in different regions. Thus, it should be necessary to exactly describe the region, in which the research has been conducted and a specific pandemic situation in this region, and selected time period.

2.2. Telephone interview

“a series of questions created ad hoc“   - why „ad hoc“?

Table 1 Please, clarify the scale of maternal/paternal education.

Table 3 Please, clarify, what the magnitudes in the table mean.

Discussion:

The discussion is divided into many short paragraphs, some of which contain even one sentence. Please join it with a more coherent text.

Page 11 “The main strengths of this study are longitudinal design“. The study was not longitudinal, only it compared two datasets within a period of six months.

The discussion mostly summarized data from the results section. I would recommend the authors to make the discussion more comprehensive.

-

Author Response

REVIEWER #1

The present manuscript deals with the important actual topic, impact of Covid-19 pandemic on children and adolescents with neuropsychiatric disorders. The authors argue that while there is already much work on the impact of a pandemic on mental health, the impact of pandemic on children with neuropsychiatric disorders has not been adequately explored. The study may provide useful findings, but it should need several improvements.

  • Abstract: I recommend not including the words "Introduction", "Material and Methods", "Results", and "Conclusions". Although abstracts usually have a similar structure, it is not common to include these designations to text.

Authors response: we thank the reviewer for the suggestion. We modified the abstract as suggested.

  • Materials and Methods: “Our observational longitudinal study…“  The study was not longitudinal, only it compared two datasets within a period of six months.

Authors response: we agree with the reviewer and we thank for the comment. We modified the manuscript replacing the definition "longitudinal study" with that suggested by the reviewer.

  • In the abstract the authors stated that “the data collected during the pandemic were compared with that collected from questionnaires administered in the six months preceding the pandemic“, however, on page 4 they stated that “The data collected during the pandemic were compared with that collected from questionnaires administered in the three months preceding the pandemic, as usual clinical practice.“ Thus, this contradiction needs a clarification.

Authors response: we apologize for our mistake. We corrected our error and specified that the comparison was made with the data collected in the six months preceding the pandemic.

  • We know that the pandemic has been going on for two years. Long lockdowns caused many mental problems. However, the data from the present study was gathered in March-April 2020, thus, just at the beginning of a pandemic. Thus, they are not very representative, which should be stated in text discussion, et least. The pandemic had several stages, alternating releases with lockdowns, in different parts of the world at different times, and the effects of the pandemic were different in different regions. Thus, it should be necessary to exactly describe the region, in which the research has been conducted and a specific pandemic situation in this region, and selected time period.

Authors response: we agree with the reviewer and we thank for the suggestion. We added a paragraph in the Results section in which we exactly described the pandemic situation in our region, at that specific time period. We have also added these issue in the limitation of the study in Discussion section.

  • Telephone interview: “a series of questions created ad hoc“   - why “ad hoc“?

Authors response: we thank the reviewer for the comment. We wanted to mean "a series of questions created by the authors specifically for the circumstance of the pandemic". We have clarified this point in the methods section.

  • Table 1 Please, clarify the scale of maternal/paternal education.

Authors response: we thank for the comment. We clarify this point as suggested by the reviewer in Table.

  • Table 3 Please, clarify, what the magnitudes in the table mean.

Authors response: we thank the reviewer for the suggestion. We clarify this point as suggested in Table 3.

  • Discussion: The discussion is divided into many short paragraphs, some of which contain even one sentence. Please join it with a more coherent text.

Authors response: we thank the reviewer for this suggestion. We modified the discussion section joining the sentences together, so that the discussion is more coherent.

  • Page 11 “The main strengths of this study are longitudinal design“. The study was not longitudinal, only it compared two datasets within a period of six months.

Authors response: we thank the reviewer for this comment. We have changed the term "longitudinal study" also in the discussion section.

  • The discussion mostly summarized data from the results section. I would recommend the authors to make the discussion more comprehensive.

Authors response: we thank the reviewer for this suggestion. We have implemented the discussion section as suggested by the reviewer.

Reviewer 2 Report

I sincerely congratulate the authors because the topic is essential. The present study is one of the best designed to answer one central question during all the COVID-19 pandemic. Having data before the pandemic and performing a good follow-up, such as this one, is outstanding. However, I feel that some changes will make the manuscript more understandable. Let me explain it:

C1. I can find several words throughout the text without any separation between them. Please, check all the manuscripts to modify this issue.

C2. Please, write down the reference number provided by the ethical committee.

C3. It is essential to provide the reliability of the answers to the questionnaires found in your research. As the authors have proposed, it is usual to say that the measures are reliable according to validation studies. But answers to an instrument could be reliable in the validation process but could not be reliable in specific research. So, please, provide the reliability in your sample.

C4. Did the authors really perform regressions? It seems they are reporting correlations. If they performed regressions, which method of introducing the independent variables (stepwise, etc) they used? We need more information.

C5. Please, provide an attrition analyisis. It seems that the authors have enough baseline data to compare the profile of participants who accepted to participate with those who did not accept.

C6. Please, provide effect sizes. See https://www.psychometrica.de/effect_size.html for the calculation of effect sizes of non-parametric statistical analyses.

C7. What means the numbers in table 5? If they are regression statistics, which one, the unstandarized or standaridzed statistic? Please, provide the t-test of each statistic or confidence interval. See APA 7th edition proposal.

C8. Please, rewrite the discussion section.  There are some lonely sentences that are really linked with the precvious and the next one, making a paragraph.

Author Response

REVIEWER #2

I sincerely congratulate the authors because the topic is essential. The present study is one of the best designed to answer one central question during all the COVID-19 pandemic. Having data before the pandemic and performing a good follow-up, such as this one, is outstanding. However, I feel that some changes will make the manuscript more understandable. Let me explain it:

C1. I can find several words throughout the text without any separation between them. Please, check all the manuscripts to modify this issue.

Authors response: we apologize for the mistake. We corrected our typos by adding spaces where they were missing.

C2. Please, write down the reference number provided by the ethical committee.

Authors response: we apologize with the reviewer for our lack. We provided the number of the ethical committee (protocol number n.0061902, approved in April 20, 2020).

C3. It is essential to provide the reliability of the answers to the questionnaires found in your research. As the authors have proposed, it is usual to say that the measures are reliable according to validation studies. But answers to an instrument could be reliable in the validation process but could not be reliable in specific research. So, please, provide the reliability in your sample.

Authors response: we thank the reviewer for the suggestion. We added the reliability of CBCL and PSI according to validation study in the method section.

C4. Did the authors really perform regressions? It seems they are reporting correlations. If they performed regressions, which method of introducing the independent variables (stepwise, etc) they used? We need more information.

Authors response: we apologize with the reviewer for the mistake. As rightly noted by the reviewer, we carried out a non-parametric correlation analysis; we specified the type of analysis in the methods section.

C5. Please, provide an attrition analysis. It seems that the authors have enough baseline data to compare the profile of participants who accepted to participate with those who did not accept.

Authors response: we thank the reviewer for this suggestion. We added the data of the subjects that refused participating in Table 1, as suggested by the author.

C6. Please, provide effect sizes. See https://www.psychometrica.de/effect_size.html for the calculation of effect sizes of non-parametric statistical analyses.

Authors response: we thank the reviewer for this suggestion. We calculated and added the effect size, as suggested in table 4.

C7. What means the numbers in table 5? If they are regression statistics, which one, the unstandarized or standaridzed statistic? Please, provide the t-test of each statistic or confidence interval. See APA 7th edition proposal.

Authors response: we thank the reviewer for this comment. We specified that these are the results of the correlation analysis.

C8. Please, rewrite the discussion section.  There are some lonely sentences that are really linked with the previous and the next one, making a paragraph.

Authors response: we thank the reviewer for this comment and suggestion. We modified and, implemented the discussion section as suggested by the reviewer.

Round 2

Reviewer 1 Report

I thank autors for adressing my comments and recommendations.